# On Limits to Flotation Cell Size

**James A. Finch \* and Yue Hua Tan**

Department of Mining and Materials Engineering, McGill University, 3610 University Street, Montreal, QC H3A 0C5, Canada
\* Correspondence: jim.finch@mcgill.ca; Tel.: +1-514-398-1452

**Abstract:** Mechanical cells have seen an exponential increase in size over the past 60 years. However, a possible size limitation due to carrying capacity constraints has been raised. Taking a Cu porphyry case, a cell sizing exercise is used to show that possible size limitation due to carrying capacity can be tested using available techniques. A range of conditions is explored that suggest a continued increase above the current maximum cell size, ca. 600 m$^3$, does not seem warranted.

**Keywords:** cell size; limitation; carrying capacity; Cu porphyry case study





## 1. Introduction

Over the past 80 years, there has been exponential growth in size (volume) of mechanical cells now represented by tank cells [1]. From less than 1 m$^3$ around 1940, maximum volumes have increased to about 600 m$^3$ today and, by extrapolation, volumes above 1000 m$^3$ would be expected by 2030. The advantage of increased cell size is a reduced number of cells for a given duty, which reduces capital and operating costs. The need to treat high tonnages of low-grade Cu porphyry deposits is one of the main drivers for this trend [2]. Recently, possible limitations on cell size have been argued [3,4]. The reasoning is that while volume increases as diameter cubed (D$^3$), a cell's cross-sectional area increases only as D$^2$, and this could lead to a (froth) carrying capacity (C$_a$, t/h·m$^2$) constraint.

Models to estimate carrying capacity (also known as the froth carry rate [5] and from a kinetic viewpoint represents zero order [3]) have been proposed [3,4,6,7]. From plant experiences, the range in C$_a$ for sulphide systems has been estimated [5,8] (for example, for roughers, it is 0.8–1.5 t/h·m$^2$).

In this paper, we show that whether cell size may be constrained by C$_a$ can be assessed using published cell sizing procedures for mechanical cells which include allowing for carrying capacity [5,8]. Sizing starts by estimating cell volume. The calculation uses target slurry feed rate with an estimate of flotation time to achieve target recovery, for example, based on lab testing with a scaling factor. This gives the required total cell volume (N·V$_{cell}$), that is, the number of cells (N) times the cell volume (V$_{cell}$). Based on practice, N for a bank is typically in the range 5–9 [2], in line with the fundamental argument for the bank to approach the plug flow transport rate that maximizes recovery for a given installed volume [1]. With N chosen, the selection of V$_{cell}$ follows, and then the target concentrate rate can be checked against the maximum rate as dictated by the carrying capacity.

## 2. Sizing Exercise

The conditions selected are based on the Cu rougher flotation data in Thompson et al. [9].

Cell size: The following conditions are assumed: dry feed rate 200,000 t/d (8333.3 t/h), solids density 3 t/m$^3$, 35% solids (*w/w*), retention time 15 min, and one open circuit bank (i.e., no recycle).

The sizing equation is as follows:

$$N \cdot V_{cell} = F \times T \times E \times P \tag{1}$$

where F is the dry solids feed rate (t/h), T is the plant retention time (h), E is the pulp expansion factor to allow for gas holdup (GH, assumed 15%), and P is the pulp volume per tonne of solids (m$^3$/t). Introducing the values:

$$E = \frac{100}{100-GH} = \frac{100}{85}$$

$$P = \frac{1}{\text{solids density}} + \frac{100}{\text{pulp \% solids}} - 1 = 2.19$$

and (to nearest whole number):

$$N \cdot V_{cell} = 5369$$

Possible options (combinations of N and V$_{cell}$) are given in Table 1, which includes cell diameter (D) and area calculated for cell aspect ratio (diameter/height) = 1, in all cases rounded to nearest whole number.

**Table 1.** Possible bank arrangement for conditions.

| Option | N | V$_{cell}$ (m$^3$) | D (m) | A$_{cell}$ (m$^2$) |
|--------|---|--------|-------|---------|
| 1 | 9 | 597 | 9 | 65 |
| 2 | 7 | 767 | 10 | 77 |
| 3 | 5 | 1074 | 11 | 97 |
| 4 | 3 | 1790 | 13 | 136 |

For the first option (N = 9), the cell size is close to the largest cell currently available, and the others represent possible future cell sizes. The first three options reflect the bank length criterion.

Carrying capacity: To check for a carrying capacity constraint, we need to compare the target concentrate rate with the maximum bank concentrate rate determined by C$_a$, which is given by C$_a$·A$_{cell}$·N (t/h). The maximum as a function of the number of cells from Table 1 for C$_a$ = 0.8–1.5 t/h·m$^2$ is shown in Figure 1.

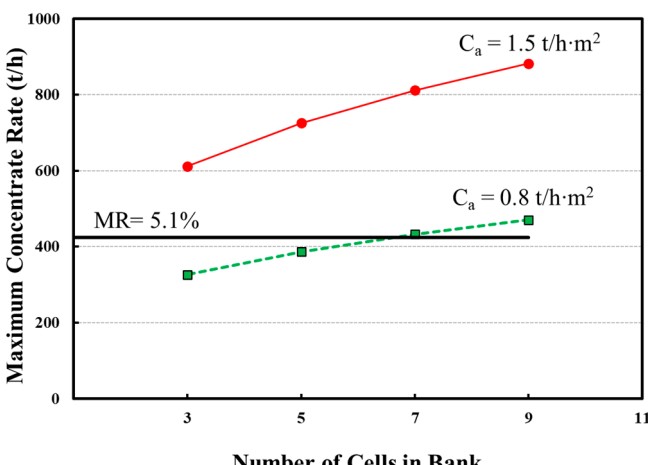

**Figure 1.** Bank maximum concentrate rate as a function of number of cells; target concentrate rate for given mass recovery (MR) (see text).

To calculate the target concentrate rate, the following conditions, approximating the rougher stage in porphyry Cu operations, are assumed: feed grade 0.3%, recovery 85%, and concentrate grade 5%, which gives 425 t/h or a mass recovery (MR) of 5.1%, which is indicated on Figure 1. The conclusion is that at the upper C$_a$ value, all bank options meet the target concentrate rate, while at the lowest C$_a$, only the first two options offer reliable capacity.

## 3. Other Capacity Constraint Measures

Carrying capacity based on area is considered the key metric [5], but other constraints that can be considered are lip loading and bubble loading.

Lip loading: Based on cell perimeter (t/h·m) from plant experience, this limit in sulphide operations is ca. 1.5 t/h·m [5]. Calculating to meet the target concentrate rate, we find a range from 1.6 t/h·m for option 1 to 3.4 t/h·m for option 4, that is, going from just acceptable to potentially problematic, but a range of launder designs is available to resolve this, and issues related to the larger froth travel distance as cell size increases [8].

Bubble loading: This can be calculated two ways: (a) per unit volume of air and (b) per unit surface area of bubbles. To calculate (a), we need the total air rate to the bank, $J_g \cdot A_{cell} \cdot N$ (m$^3$/h), where $J_g$ is the gas's superficial velocity (m/h), and for (b), we need total bubble surface area per unit time, $S_b \cdot A_{cell} \cdot N$, where $S_b$ is the bubble surface area flux (1/h). Taking conservative values from [1], we select $J_g$ = 1 cm/s (36 m/h) and $S_b$ = 40 1/s (144,000 1/h). The results for both loadings, converted to the units common in the literature, are given in Table 2 (the range is for option 1 to option 4).

**Table 2.** Bubble loadings from literature compared with estimate for present example.

| Reference | Bubble Loading | | Comments | |
|---|---|---|---|---|
| | Mass Per Unit Surface Area (mg/mm$^2$) | Mass Per Unit Volume (g/L) | | |
| Bradshaw and O'Connor [10] | 0.019(±0.06)–0.107(±0.35) | — | | Microflotation cell |
| Chegeni et al. [11] | 0.0188–0.0434 | — | Measured in | Lab column |
| Eskanlou et al. [12] | 0.0002–0.032 | — | | |
| Eskanlou et al. [13] | 0.0129–0.0791 | — | | |
| Ostadrahimi et al. [14] | 0.0027–0.0043 | 8.40–26.34 | | Industrial setting |
| Yianatos and Contreras [15] | | 24–70 | | |
| Yianatos et al. [16] | — | 26.8–51.0 | Calculated in | Industrial setting |
| Finch and Dobby [6] | — | 32–78 | | |
| Present work | 0.0050–0.0072 | 20–29 | Estimated in | |

The literature ranges are wide, partly reflecting mineral density; for example, in Eskanlou et al. [13], the range in (b) goes from the low value with quartz to the high value with galena. The values are not necessarily the maximum, but regardless, the values for the present work fall within the published ranges, even for the largest cell option (option 4). Finch and Dobby [6] considered using (b) (given symbol $C_g$) but compared to $C_a$, $C_g$ was a stronger function of $J_g$. The comparative lack of effect on $C_a$ was attributed to bubble size ($D_b$) being a function of $J_g$ such that an increase in $J_g$ also increased $D_b$, thus offsetting the anticipated increase in bubble surface area flux, $S_b$, whereas no such offsetting occurs in the case of $C_g$. Ostadrahimi et al. [14] also remarked on $C_g$ being a function of $J_g$.

## 4. Discussion

Without altering geometry substantially, increasing cell size results in a lesser increase in surface area relative to the increase in volume, which impacts carrying capacity and lip loading, and if the gas superficial rate and bubble surface area flux are specified, it also impacts bubble loading. This exercise is intended to show that at least a preliminary examination of all these possible constraints on cell size can be assessed using available techniques.

The most important constraint is considered to be carrying capacity [5,8]. It is best determined experimentally [6], but failing that, the $C_a$ range found by experience has an advantage over model estimates which require a variety of assumptions. One feature shared by the models is that $C_a$ increases as particle size increases and, since porphyry Cu plants employ coarse primary grinds to reduce energy costs, this favours the upper end of the $C_a$ range (1.5 t/h·m$^2$); that is, all bank options would suffice for the conditions considered. Moreover, unless there is a compelling reason for having a single bank with fewer than nine cells, then option 1 with the currently available maximum tank size (ca. 600 m$^3$) meets the requirements; that is, there may be no need for continued growth in cell size. If two

banks were used, and there are operational advantages to having a parallel line, then the calculations show that the 600 $m^3$ cell would suffice with as few as five cells, even at the low-end $C_a$.

Possible conditions that could be considered are obviously numerous. For example, concentrate rate could be increased by lower concentrate grade or higher feed tonnage, but the increase could be handled by a second, parallel bank. Recycle could be included, but while this complicates the calculation, testing for $C_a$ remains the same.

Calculations have been based on the bank, which may imply each cell has to have equal (balanced) mass pull. There is a case for such balanced bank operation to optimize performance [17], but operations often pull the first cell hard, raising concern that this cell reaches carrying capacity, even requiring a tailored launder design [8]. This should not be an issue, however, provided the bank concentrate rate is achievable; that is, the downstream cells 'pick up the slack'.

If instead of Cu it is a Zn deposit, then a possible situation is feed grades 10% and rougher recovery 80% at concentrate grade 40%, which gives a mass pull of 20%, which would exceed any option in Figure 1. However, in the Zn case, feed rates are much lower than 200,000 t/d, closer to 10,000 t/d. For the given conditions (10% feed grade, etc.), targeting a concentrate rate of 300 t/h, which is comfortably met by all options, even at the low end of the $C_a$ range, means a feed rate up to 36,000 t/d could be processed.

In any sulphide plant, mass recoveries are high (>50%) in the cleaning stage(s) and carrying capacity can be a factor. For example, Finch and Dobby [6] encountered a $C_a$ constraint testing a column which was resolved by placing units in series, that is, effectively a bank of columns. Feed rates to cleaners, however, are low compared to the rougher (in this Cu porphyry example, 425 t/h compared to 8333.3 t/h), and there does not seem to be much incentive to explore larger cells coming from cleaning duties.

### 5. Conclusions

It is shown that a potential carrying capacity ($C_a$) constraint on mechanical cell size can be examined using standard cell selection methodology. Based on a Cu porphyry case and related observations, there does not appear to be strong incentive for a further increase in cell size beyond the current maximum of ca. 600 $m^3$.

**Author Contributions:** Conceptualization, J.A.F. and Y.H.T.; methodology, J.A.F.; validation, J.A.F. and Y.H.T.; formal analysis, J.A.F.; data curation, J.A.F. and Y.T.; writing—original draft preparation, J.A.F.; writing—review and editing, Y.H.T.; visualization, Y.H.T.; supervision, J.A.F. All authors have read and agreed to the published version of the manuscript.

**Funding:** This research received no external funding.

**Data Availability Statement:** The data presented in this study are available on request from the corresponding author.

**Conflicts of Interest:** The authors declare no conflict of interest.

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
