# Peer review of "On Limits to Flotation Cell Size"

_minerals, doi:10.3390/min13030411_

Round 1

Reviewer 1 Report

This paper will be of interest to flotation practitioners and is well worth publishing. My comments, which are of relatively minor importance, are as follows:

Line 26. Reference [5] should be deleted. The ordinary reader will have no access to a paper presented at a conference with limited attendance.

Perhaps use could be made of Espinosa-Gomez, R.; Finch, J.A.; Yianatos, J.B.; Dobby, G.S. Flotation column carrying capacity: Particle size and density effects. Miner. Eng. 19881, 77–79.

Carrying capacity values in mechanical cells have been reported by Yianatos and Contreras. This paper should be referenced: 

On the Carrying On the Carrying Capacity Limitation in Large Flotation Cells

  • October 2010
  • Canadian Metallurgical Quarterly 49(4):345-352

DOI:10.1179/000844310795937578 Authors:   Juan Yianatos

  • Universidad Técnica Federico Santa María

      Felipe Andres Contreras

  • Anglo American

Line 44, the reference to Wood [9] should be deleted. The total volume is self evident and equation (1) does not require a reference, especially to one that requires some effort to locate.

Lines 45 to 53 are poorly presented, perhaps due to the manuscript handling by the publisher. Please present in a format that is easy to understand.

One of the issues not discussed is the role of froth "crowding". Crowding is often used when the mass of floatable material is so low that it is not possible to generate a stable froth over the total area of a large flotation cell. What does crowding do to the carrying capacity? Would the installation of crowders or special launders to increase the lip length to area ratio be of benefit? Some comments are required

Author Response

  1. I would prefer to keep reference 5. One can find it by searching for ‘Selection of Mechanical Flotation Equipment, Presented at METCON 2013’ which takes you right to the paper. I like the paper as it is one of the first to provide the estimates of Ca that industry use and thus avoided trying to estimate from models.
  2. The Espinosa paper certainly fits, but the Finch/Dobby book provides the same data and extends to assess carrying capacity based on area (Ca) vs that based on gas volumetric flow (Cg), discussed in lines ca. 90-93, and showing Ca was less dependent on gas rate.
  3. We have added this Yianatos reference to Table 2 giving another estimate of bubble load. We do recognize that not all bubble load estimates we are aware of are included, partly because they all fall within ranges similar to those already in the table.
  4. We have removed the Wood reference and re-written lines 45-52 to try to improve the flow.
  5. Froth crowding seems a separate issue. If crowders are needed to build sufficient particle mass to support a froth this would seem we are far from the carrying capacity (quite the contrary in fact). We do not think this topic fits the paper.

Reviewer 2 Report

This is a very useful paper particularly for those readers who are not familiar with the flotation cell technology. With the target audience in mind, I would only have a couple of suggestions for authors consideration:

1) The authors should consider additional comments in their discussion on choosing, say, one very large cell over a bank of smaller cells, how these decisions are made based on ore properties and kinetic criteria. Explain the  term "bank length criterion".

2) It would also be helpful to comment on some hydrodynamic aspects. How does the power-to-volume ratio change with cell size, and how this trend looks like for the largest cells.

Otherwise, the paper is recommended for publication in the journal.

Author Response

  1. On lines 32-34 we that common practice is 5-9 cells in series to form a bank (or row or line). The argument is that while single cells approach fully mixed flow by placing them in series the transport approaches plug flow, and this increases recovery for a given installed cell volume. The series arrangement is widely applied and needs no further justification. It is rare that a single cell is used.
  2. We agree from a cell manufacturer’s perspective power to volume ratio is important, but such discussion would be outside the scope of this paper.

Reviewer 3 Report

This manuscript aims to discuss the limits of flotation cell size. The topic is interesting for the flotation community. Cell size is something that both flotation equipment manufacturers, as well as processing plants, are interested in. However, the manuscript is too short, and it is not ready to be considered for publication as a journal paper. The conclusion has been also made from the results of one case study and it cannot be generalized. The authors can expand these data a by examining some more case studies if available. The manuscript in its current form is rather suitable for a Technical Note. 

Author Response

The main point is the length, which is felt to be more suited to technical note. We have no objection to such designation if the editor wishes

Round 2

Reviewer 3 Report

I have no further comments